# Surprisingly Simple: Large Language Models are Zero-Shot Feature Extractors for Tabular and Text Data

## Abstract

Large Language Models (LLMs) have demonstrated remarkable capabilities across diverse tasks, yet their application to tabular data prediction remains relatively underexplored. This is partly due to the fact that recent LLMs are autoregressive models, generating text outputs. Converting tabular data into text, and vice versa, is not straightforward, making direct application of LLMs to complex tabular prediction difficult. Although previous works have utilized pre-trained embedding models like BERT and its variants for fine-tuning on tabular tasks, the potential of autoregressive LLMs for tabular prediction has been explored only on a limited scale and with simpler datasets. In this paper, we propose **Z**ero-shot **E**ncoding for **T**abular data with **LLM**s (ZET-LLM), a surprisingly simple yet effective approach that leverages pre-trained LLMs as zero-shot feature extractors for tabular prediction tasks. To adapt autoregressive LLMs for this purpose, we replace autoregressive masking with bidirectional attention to treat them as feature embedding models. To address the challenge of encoding high-dimensional complex tabular data with LLMs' limited token lengths, we introduce a feature-wise serialization, where each feature is represented as a single token, and the resulting tokens are combined into a unified sample representation. Additionally, we apply missing value masking to handle missing data, a common issue in complex tabular datasets. We demonstrate that LLMs can serve as powerful zero-shot feature extractors without the need for fine-tuning, extensive data pre-processing, or task-specific instructions. Our method enables LLMs to process both structured tabular data and unstructured text data simultaneously, offering a unique advantage over traditional models. Extensive experiments on complex tabular datasets show that our approach outperforms state-of-the-art methods across binary classification, multi-class classification, and regression tasks.

## 1 Introduction

Understanding complex tabular data accompanied by natural language text presents a significant challenge but is crucial across a wide range of applications. One prominent example is medical data, particularly electronic health records (EHRs), which are a rich and complex source of patient information. EHRs capture longitudinal health data, encompassing both structured fields and unstructured content, such as clinical notes. This combination introduces complexities beyond those found in typical tabular datasets, including high interconnectivity between variables, the presence of missing or incomplete data, and often noisy signals. Effectively analyzing such datasets is critical not only in the medical domain but also in other fields such as financial systems and e-commerce, where intricate data involving both tabular and textual information is commonly generated.

Prediction models for tabular data have been extensively studied, with gradient-boosted decision trees (GBDT), such as XGBoost Chen & Guestrin (2016), LightGBM Ke et al. (2017), and CatBoost Prokhorenkova et al. (2018), remaining highly competitive despite the emergence of newer approaches. On the other hand, deep learning methods for tabular data prediction, which encompass various architectures, e.g., TabNet Arik & Pfister (2021), NODE Popov et al. (2022), SAINT Somepalli et al. (2021), have also gained traction by modeling complex feature interactions and leveraging self-supervised learning strategies. However, despite the significant advancements

in deep learning, GBDT algorithms still perform favorably against these models on several scenarios Shwartz-Ziv & Armon (2022); Gorishniy et al. (2021); Grinsztajn et al. (2022). As of now, there are no clear winners between deep learning and tree-based methods, as each presents unique strengths and limitations depending on the nature of the data and task.

Despite the widespread success of deep learning in fields such as computer vision and NLP, it has not yet outperformed traditional methods in tabular data modeling. This is largely due to the inherent complexity of tabular data, which often combines diverse feature types—such as categorical, numerical, and textual data—making it challenging for deep learning models to process effectively. Additionally, real-world tabular datasets tend to be sparse, with imbalanced classes and missing values, particularly in domains like healthcare and fraud detection, further complicating model training. Another key challenge is the heavy reliance on pre-processing Borisov et al. (2022b). Proper scaling, encoding, and imputation are critical to prevent information loss or new problems such as multicollinearity. Furthermore, the interdependent nature of features in tabular data can be difficult for deep learning models to capture, as correlations between variables can significantly impact predictions. Unlike tasks involving images or audio, tabular data lacks clear spatial or temporal structures, making it harder for deep learning models to infer meaningful relationships without extensive data and computation.

Leveraging pre-trained language models for tabular prediction tasks is a promising direction due to several advantages. First, tabular data can be easily converted into a text format using simple serialization techniques, such as "[feature] is [value]," which simplifies the pre-processing of heterogeneous data types. Instead of applying complex, specialized pre-processing to each feature type, this approach unifies all features into a consistent text format. Second, by converting heterogeneous features into the same text domain, language models overcome the challenge of understanding diverse data types, making it easier to process categorical, numerical, and textual data uniformly. Additionally, large, complex datasets like electronic health records (EHRs) often contain missing data, which poses significant challenges for traditional models. Solutions such as data imputation or filling in missing values are empirical and can introduce noise. In contrast, language models can handle missing data by simply masking missing tokens. Finally, pre-trained language models have the added advantage of incorporating context. While traditional models tend to overlook the context of feature names and their relationships, language models can understand and use this context to improve predictions.

Given the aforementioned advantages, several efforts have been made to leverage pre-trained language models (PLMs) for tabular data prediction. Most of these approaches Liu et al. (2022); Yan et al. (2024) involve fine-tuning embedding-based PLMs like BERT Devlin (2018) by attaching task-specific heads to adapt them for tabular prediction tasks. More recently, there has been growing interest in using large language models (LLMs), such as GPT and its variants Brown (2020); Gao et al. (2020), for tabular prediction. Given the success of LLMs on various applications, it is natural to consider that LLMs might potentially outperform PLMs like BERT in tabular data tasks. While some works have attempted to fine-tune LLMs on textualized tabular data Dinh et al. (2022); Hegselmann et al. (2023), this approach is computationally expensive due to the size and complexity of LLMs. Additionally, fine-tuning introduces various hyperparameters and requires complex scheduling strategies.

In this paper, we introduce ZET-LLM (**Z**ero-shot **E**ncoding for **T**abular data with **LLM**s). To adapt LLMs as tabular feature extractors, we propose several key modifications. First, while LLMs typically predict next tokens, we convert them into embedding models by replacing autoregressive masks with bidirectional attention, inspired by LLM2Vec BehnamGhader et al. (2024). This enables LLMs to output embeddings rather than generating text. Second, we explore different serialization strategies and find that feature-wise serialization outperforms the widely-used sample-wise serialization. The feature tokens are then aggregated by a Feature Integrator, which is trained alongside the prediction layer for task-specific adaptation.

ZET-LLM is computationally efficient as it fine-tunes only the task-specific modules, eliminating the need for post-processing steps typically required by autoregressive LLMs. Moreover, unlike traditional tabular models, ZET-LLM naturally integrates both tabular data and text descriptions, significantly improving performance on complex tabular tasks. In summary, ZET-LLM achieves state-of-the-art performance on challenging tabular prediction tasks while reducing preprocessing and feature engineering requirements.

Our contributions are as follows:

- We propose ZET-LLM, a simple method that leverages LLMs as feature extractors for tabular data without the need for fine-tuning, preprocessing, or task-specific instructions.

- Our method encodes both structured tabular data and unstructured text simultaneously.

- Extensive experiments validate the effectiveness and robustness of our approach on complex tabular prediction tasks.

- Ablation studies demonstrate the flexibility of our method across a range of pre-trained LLMs, highlighting its versatility.

## 2 RELATED WORK

### 2.1 TABULAR PREDICTION BEFORE LANGUAGE MODELS

Tabular data prediction has long been dominated by tree-based models, with methods such as Gradient-Boosted Decision Trees (GBDT) and its variants, including XGBoost Chen & Guestrin (2016), LightGBM Ke et al. (2017), and CatBoost Prokhorenkova et al. (2018), achieving strong performance across a variety of tasks. Later, deep learning approaches gained attention for tabular prediction tasks. Methods like SAINT Somepalli et al. (2021), TabNet Arik & Pfister (2021), and NODE Popov et al. (2022) focus on learning feature representations end-to-end, using self-supervised learning and attention mechanisms to model complex feature interactions. Furthermore, models such as TabTransformer Huang et al. (2020), SAINT Somepalli et al. (2021), and TransTab Wang & Sun (2022) utilize transformer architectures to capture dependencies between features. Some works combine deep learning and decision trees to combine the strengths of both approaches, as seen in methods like DeepGBM Ke et al. (2019) and BGNN Ivanov & Prokhorenkova (2021). For a comprehensive overview of deep learning techniques applied to tabular data, refer to Borisov et al. (2022a). Gorishniy et al. Gorishniy et al. (2021) conducted extensive experiments comparing various architectures and methods for tabular prediction but found no clear winner among the different categories of models. While deep learning approaches have demonstrated potential in certain scenarios, they still often struggle to consistently outperform traditional tree-based models on tabular datasets Shwartz-Ziv & Armon (2022); Grinsztajn et al. (2022).

### 2.2 TABULAR PREDICTION WITH LANGUAGE MODELS

In many cases, pre-trained language models (PLMs) are initially trained on large textual corpora are later fine-tuned for tabular prediction tasks. Embedding-based PLMs, particularly BERT Devlin (2018) and its variants, have been adapted for tabular prediction. For instance, TP-BERTa Yan et al. (2024) uses RoBERTa Liu (2019) as a pre-trained language model and fine-tunes it with task-specific loss on a wide variety of tabular datasets to capture the structure of tabular data. Similarly, PTab Liu et al. (2022) fine-tunes BERT on tabular datasets using both task-specific loss and masked token prediction on serialized tabular data. In domain-specific applications, models like CTRL Li et al. (2023) fine-tune RoBERTa on financial tabular data, while MediTab Wang et al. (2024) fine-tunes BioBERT Lee et al. (2020) for medical Electronic Health Record (EHR) datasets, making them highly specialized for their respective fields.

Recently, large language models (LLMs) with more than 1B parameter sizes have been widely used for tabular data. Unlike embedding-based PLMs, LLMs are autoregressive models such as GPT-J Gao et al. (2020), GPT-3 Brown (2020), LLaMA2 Touvron et al. (2023b), and T0 Sanh et al. (2022). One approach to leveraging LLMs for tabular prediction is to treat them as agents that generate text-based responses. Several methods attempt to fine-tune autoregressive LLMs for tabular prediction. LIFT Dinh et al. (2022), TabLLM Hegselmann et al. (2023), and GTL Zhang et al. (2023a) are notable examples where autoregressive LLMs are fine-tuned on serialized tabular data. These methods serialize the tabular data into sentence-like formats and mask certain parts of the input to predict missing values or target labels. However, fine-tuning large language models is computationally expensive and involves hyperparameters with complex scheduling strategies.

Another line of work explores zero-shot inference using LLMs, which leverage pre-trained models without any task-specific fine-tuning. These methods rely on techniques like in-context learn-

ing Brown (2020), instruction prompting Sanh et al. (2022), and chain-of-thought (CoT) reasoning Wei et al. (2022). In zero-shot settings, models like Tablet Slack & Singh (2023) use task-specific instructions to boost performance, while SummaryBoost Manikandan et al. (2023) generates natural text descriptions of tabular data and summarizes them for few-shot in-context learning. Despite these attempts, zero-shot approaches remain limited by the pre-trained knowledge of the LLMs, as they do not undergo fine-tuning for the specific domain or task. This makes them less effective in complex, domain-specific contexts like medical or finance, where precise understanding of the relationships between features is critical. Furthermore, instruction design for complex tabular data can be challenging, and the few examples used in in-context learning may be insufficient to fully capture intricate feature interactions, particularly when dealing with large-scale tabular datasets that frequently exceed the token limits of the model.

In this paper, we address these limitations by proposing ZET-LLM, which combines the strengths of both fine-tuning-based and zero-shot inference LLMs. ZET-LLM avoids the high computational cost associated with fine-tuning the entire LLM by only fine-tuning the task-specific layers for downstream tasks. This makes it more lightweight while still effectively adapting to the target task, a capability that zero-shot methods cannot achieve.

## 3 METHOD

In this section, we describe how pre-trained large language models (LLMs) can be employed as zero-shot feature extractors for tabular prediction tasks. Our framework consists of four key stages: 1) feature-wise tabular-to-text serialization, 2) Missing feature token masking, 3) Feature aggregation and encoding using attention blocks, and 4) Prediction. The overview of our framework is illustrated in Figure 1.

**Tabular-to-Text Serialization** Let us define tabular data as a set of feature-name and value pairs $(k, v)$. Each sample in the dataset is represented as a set of such pairs $x = \{(k_i, v_i) | i = 1 \ldots, n\}$ where n denotes the number of features. The task is to predict a label for classification or a continuous value for regression. To leverage LLMs for feature extraction, we first convert the tabular data into a text format. We adopt a straightforward approach for tabular-to-text serialization, similar to previous work Hegselmann et al. (2023); Dinh et al. (2022); Jaitly et al. (2023), where each feature $(k_i, v_i)$ in the tabular data is serialized into a simple sentence format as follows:

$$t_i = \text{"}[k_i] \text{ is } [v_i]\text{"}. \tag{1}$$

By this process, each tabular data sample is transformed into a set of descriptive sentences. Prior methods Hegselmann et al. (2023); Dinh et al. (2022); Jaitly et al. (2023) aggregate these sentences into a paragraph that encodes the entire feature set into a single token using LLM. However, we find this strategy suboptimal for tabular prediction tasks due to several reasons.

First, tabular data is inherently order invariant, meaning the sequence of features should not impact the prediction. When sentences are concatenated into paragraphs, the ordering of features may result in varying predictions, which conflicts with the nature of tabular data. Second, LLMs often attend more strongly to tokens at the end of a sequence. This bias can cause certain features to be weighted more heavily simply because they appear later in the serialized text, which may result in suboptimal predictions. Third, when tabular data contains numerous features, encoding all of them into a single token burdens the LLM with a large set of information. This issue is particularly problematic if the features require domain-specific knowledge to interpret. Lastly, as the number of features grows, so does the length of the serialized text. If the number of tokens exceeds the LLM's token limit, important features may be truncated, reducing the model's ability to make accurate predictions.

To address these limitations, we propose encoding each feature as a separate token rather than representing the entire sample as a single token. By treating each feature individually, we ensure that the model attends to all features more uniformly, preserving the order-invariance property and alleviating the aforementioned issues. Our experiments, detailed in the ablation study section, demonstrate the effectiveness of this approach in comparison to the prior serialization method.

**Encoding Using LLM** Since LLMs are autoregressive, they are not directly suited for text embedding as they generate tokens sequentially based on prior tokens. Inspired by

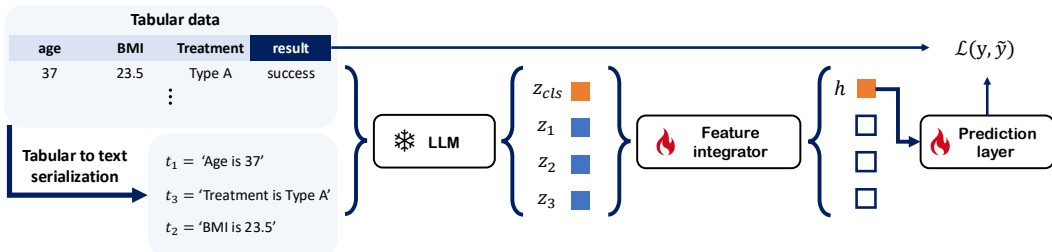

Figure 1: Overview of the proposed framework. 1) Tabular data is serialized into text sentences. 2) Each sentence is encoded into feature tokens using a pre-trained LLM. 3) The Feature Integrator aggregates the feature tokens to generate a representation for the entire data sample. 4) The sample representation is passed to the prediction layer. 5) The final prediction is compared with the ground truth label, and the loss is computed.

LLM2Vec BehnamGhader et al. (2024), we modify the architecture by replacing the LLM's autoregressive masking with bidirectional attention. This enables the model to consider the entire context of a sentence at once, rather than generating tokens one by one. For each serialized feature sentence $t_i$, the LLM encodes it into a sequence of token embeddings. To represent each sentence as a single token, mean pooling is applied across the tokens within the sentence Muennighoff (2022). This process generates a feature token $z_i$ for each serialized sentence $t_i$, and the resulting set of feature tokens for a data sample is represented as:

$$\mathbf{z} = \{z_i | i = 1 \ldots, n\}, \quad z_i = \mathcal{P}(f_{LLM}(t_i)) \tag{2}$$

where $f_{LLM}$ denotes a pre-trained LLM and $\mathcal{P}$ denotes a pooling operation. This approach enables us to generate fixed-length embeddings for each feature.

**Missing Value Masking** In complex tabular data, such as in the medical domain, it is common to encounter missing values. These missing values introduce noise into the data, often deteriorating the performance of models trained on such datasets. To mitigate the adverse effects of missing values, we apply missing value masking before aggregating the feature tokens. Specifically, during the encoding process, features with missing values are ignored, and only features with valid values are retained for encoding. This ensures that the resulting feature tokens represent only the available data, reducing the influence of missing information on the model's performance.

**Encoded Feature Token Aggregation** Once the features have been serialized and missing values masked, the LLM acts as a feature token extractor, generating a token for each feature that captures its semantic representation. However, to make predictions, these feature tokens must be aggregated into a single representation of the entire data sample. To achieve this, we employ transformer blocks as the Feature Integrator to combine the individual feature tokens. A trainable class $z_{cls}$ token with random initialization alongside the encoded feature tokens $\{z_i\}$ is also added. The Feature Integrator learns to merge the information from each feature token into the class token, representing the full data sample as follows:

$$h = f_{FI}(\mathbf{z}, z_{cls}) \tag{3}$$

Where $f_{FI}$ represents the Feature Integrator.

In our framework, we find that a shallow architecture, specifically a single transformer block, is sufficient for this task. Decoupling feature encoding from feature aggregation offers several key advantages. First, by encoding each feature separately, the model captures the contextual information of each feature independently. This removes concerns related to the order of feature-value pairs, which can be problematic in LLM-based approaches where the sequence of features or the model's tendency to prioritize later tokens might skew the representation.

Additionally, this approach allows for the efficient utilization of features. The transformer-based Feature Integrator is trained to combine the encoded feature tokens, ensuring that all features contribute appropriately to the downstream task. This divide-and-conquer strategy allows the pre-trained LLM to focus solely on embedding contextual information for each feature, while the Feature Integrator handles the task of combining them effectively.

Table 1: Summary of Datasets.

| Datasets | Task | Domain | Number of Samples | | | Number of Features | | | Missing Rates | Notes |
|---|---|---|---|---|---|---|---|---|---|---|
| | | | Train | Validation | Test | Numerical | Categorical | Textual | | |
| SAD | binary classification | medicine | 11696 | 1462 | 1462 | 35 | 2 | 0 | 0.007 | — |
| Mortality | binary classification | medicine | 16911 | 2114 | 2114 | 15 | 3 | 0 | 0.601 | missing values |
| Decompensation | binary classification | medicine | 16000 | 2000 | 2000 | 15 | 3 | 0 | 0.691 | missing values |
| Respiration | binary classification | medicine | 12208 | 1526 | 1527 | 15 | 3 | 0 | 0.691 | missing values |
| Sepsis | binary classification | medicine | 12208 | 1526 | 1527 | 15 | 3 | 0 | 0.691 | missing values |
| Shock | binary classification | medicine | 12208 | 1526 | 1527 | 15 | 3 | 0 | 0.691 | missing values |
| IVF Pregnancy | binary classification | medicine | 1347 | 175 | 175 | 4 | 3 | 1 | 0.013 | domain knowledge |
| Fake Job | binary classification | advertisement | 1370 | 174 | 188 | 2 | 8 | 5 | 0.330 | missing values |
| Thumbs-Up | 5 classification | rating | 154665 | 43710 | 39310 | 2 | 5 | 1 | 0.000 | — |
| Healthcare | 3 classification | medicine | 44400 | 5550 | 5550 | 3 | 9 | 0 | 0.000 | — |
| Body Performance | 4 classification | medicine | 10714 | 1339 | 1340 | 10 | 1 | 0 | 0.000 | — |
| Sales | regression | marketing | 7840 | 980 | 980 | 1 | 15 | 0 | 0.000 | — |
| Stock | regression | marketing | 3664 | 458 | 459 | 5 | 0 | 0 | 0.000 | — |
| Air Quality | regression | rating | 3832 | 479 | 479 | 5 | 12 | 0 | 0.000 | — |
| Employee Tenure | regression | career | 1176 | 147 | 147 | 24 | 8 | 0 | 0.000 | — |

Finally, the approach enhances interpretability through the use of attention scores. The attention mechanism within the feature aggregation process assigns scores to the individually encoded features, which contribute to the class token representing the entire data sample. These attention scores directly indicate feature importance, providing valuable insights into which features are most relevant for the downstream task. This interpretability facilitates further analysis of the dataset and the task itself, offering a more transparent model behavior.

**Prediction Layer** The aggregated feature tokens are passed through a task-specific prediction layer to produce the output for the target task. We use a Multi-Layer Perceptron (MLP) as the prediction module. For classification tasks, the MLP is trained with cross-entropy loss to predict the appropriate class label. For regression tasks, the MLP is trained with L2 loss to predict continuous values. Our method directly produces task-specific outputs, such as class labels or numerical values, without requiring post-processing. Unlike autoregressive LLMs that generate text needing conversion into labels or values, our approach simplifies the workflow, improving both efficiency and accuracy without potential post-processing errors.

# 4 EXPERIMENTS

## 4.1 DATASETS

We evaluate our method on a variety of datasets to demonstrate its capacity in different scenarios. Detailed information on all datasets is shown in Table 1.

Binary classification is the most common task in tabular prediction. Our binary classification tasks are mainly from MIMIC-III Johnson et al. (2016) and MIMIC-IV Johnson et al. (2023), which are real-world databases containing EHRs of patients admitted to the Beth Israel Deaconess Medical Center. They contain comprehensive information about hospitalizations and ICU stays, such as medication administration and laboratory measurements. We refer to past works to build prediction tasks from these data. Following Zhang et al. (2023b), we evaluate tabular prediction models on **Sepsis-Associated Delirium (SAD)** prediction task, which is a complex syndrome associated with poor prognosis and long-term cognitive dysfunction. Following Harutyunyan et al. (2019), we evaluate tabular prediction models on predicting the occurrence of **Mortality**, or in-hospital mortality; **Decompensation**, or the rapid deterioration of patients' systems; **Respiration**, or respiratory failure; **Sepsis**, or the body's overreaction to infection and injury; and **Shock** amongst patients in an adult ICU. Additionally, we also adopt **In-Vitro Fertilization (IVF) Pregnancy**, which is a clinical dataset Kim et al. (2024) from the Tel Aviv Sourasky Medical Center to predict this outcome from EHR data and doctors' comments. As for the general task, we take **Fake Job** Cohen (2020), predicting whether job postings are real or fake.

For the multi-class classification tasks, we select three datasets. The first is **Thumbs Up** created by PPrior Fereidouni et al. (2022), predicting the rating of Google Play reviews. From Kaggle, we take **Healthcare** and **Body Performance**, predicting the health level of human bodies.

For the regression tasks, we select relatively large size datasets from the regression tasks of TP-BERTa Yan et al. (2024). Specifically, **Sales** predicts the sales of various products in a supermarket; **Stock** predicts the stock prices of Netflix over the past 10 years; **Air Quality** forecasts the air quality

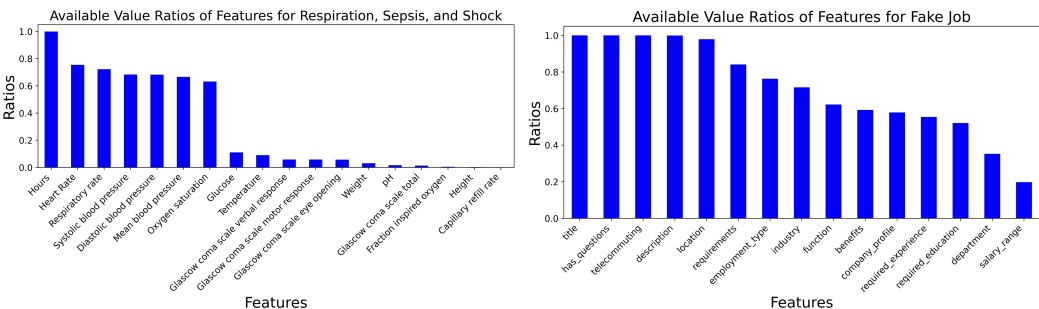

Figure 2: Left: available value ratios across different features in Respiration, Sepsis, and Shock. Right: available value ratios across different features in Fake Job.

| Company Profile | Requirements | Last Visit Comments |
|---|---|---|
| Our passion for improving quality of life through geography is at the heart of everything we do. | EDUCATION: Bachelor's or Master's in GIS, business administration, or a related field, or equivalent work experience, depending on position | Approximately a week after the embryo transfer, the patient experienced swelling and abdominal pain. The abdomen was slightly enlarged and soft, with no signs of peritoneal irritation. |
| A synthetic example of Fake Job | | A synthetic example of IVF Pregnancy |

Figure 3: Left: a synthetic example of two textual features in Fake Job. Right: a synthetic example of a textual feature in IVF Pregnancy.

across different regions in Minneapolis; and **Employee Tenure** M S et al. (2023) predicts the tenure of employees at IBM.

These datasets are primarily intended to test our model's performance in three aspects:

**Missing Values**    Missing values are a common issue in tabular data, particularly in medical datasets like EHRs. Given their prevalence, evaluating models on data with missing values is critical for tabular predictions. Many of the datasets we use contain a significant proportion of missing values, with some having high missing rates. Table 1 presents the overall missing value rates for each dataset, where the missing value rate is defined as the average rate of missing data across all features. For further illustration, Figure 2 provides examples showing the available value ratios across different features.

**Textual Features**    To demonstrate our advantage in handling unstructured texts, we select datasets containing entire paragraphs of textual features. Figure 3 shows synthetic examples of textual features. These textual features are usually a whole paragraph and describe related information about the samples. Since traditional tabular prediction models cannot process such information, we categorize the features into numerical, categorical, and textual features, as summarized in Table 1. Initially, we experiment using only the numerical and categorical features to ensure a fair comparison with previous tabular models. We then incorporate the textual features to demonstrate the importance of integrating unstructured text with other tabular features.

**Domain Knowledge**    Typically, LLMs require fine-tuning to incorporate relevant domain knowledge for specific downstream tasks. However, since our approach does not involve fine-tuning the LLM, we aim to evaluate its effectiveness in scenarios where domain-specific understanding is essential. For example, as illustrated in Figure 3, the IVF Pregnancy dataset includes doctors' comments on patients' health conditions and test results. These comments often contain specialized medical terminology that is uncommon in general language corpora. This makes IVF Pregnancy an ideal dataset to evaluate our zero-shot method's ability to handle domain-specific language without prior fine-tuning, highlighting the model's adaptability in tasks requiring expert knowledge.

Table 2: Results across datasets for different tasks. "Texts" refers to the results after incorporating textual features. ↑ indicates that higher values are better, while ↓ indicates that lower values are preferred. Among the results without textual features, the best performances are highlighted in bold, and the second-best performances are underlined.

| Methods | Binary Classification (AUC) ↑ | | | | | | | | Mean ↑ |
|---|---|---|---|---|---|---|---|---|---|
| | SAD | Mortality | Decompensation | Respiration | Sepsis | Shock | IVF Pregnancy | Fake Job | |
| XGBoost | 0.835 ± 0.000 | 0.655 ± 0.000 | 0.921 ± 0.000 | 0.763 ± 0.000 | 0.660 ± 0.000 | 0.735 ± 0.000 | 0.526 ± 0.000 | 0.476 ± 0.000 | 0.696 |
| RF | 0.846 ± 0.003 | 0.653 ± 0.010 | 0.776 ± 0.049 | **0.769 ± 0.004** | 0.668 ± 0.006 | 0.725 ± 0.007 | 0.522 ± 0.019 | 0.594 ± 0.005 | 0.694 |
| SVM | 0.805 ± 0.000 | 0.609 ± 0.000 | 0.793 ± 0.000 | 0.758 ± 0.000 | 0.622 ± 0.000 | 0.494 ± 0.042 | 0.500 ± 0.000 | 0.671 ± 0.001 | 0.657 |
| Logistics | 0.816 ± 0.000 | 0.627 ± 0.000 | 0.861 ± 0.000 | 0.733 ± 0.000 | 0.647 ± 0.000 | 0.701 ± 0.000 | 0.603 ± 0.000 | 0.726 ± 0.000 | 0.714 |
| KNN | 0.715 ± 0.000 | 0.579 ± 0.000 | 0.781 ± 0.000 | 0.695 ± 0.000 | 0.609 ± 0.000 | 0.673 ± 0.000 | 0.499 ± 0.000 | 0.547 ± 0.000 | 0.637 |
| Bayes | 0.763 ± 0.000 | 0.568 ± 0.000 | 0.845 ± 0.000 | 0.661 ± 0.000 | 0.598 ± 0.000 | 0.604 ± 0.000 | **0.615 ± 0.000** | 0.672 ± 0.000 | 0.666 |
| MLP | 0.807 ± 0.011 | 0.657 ± 0.014 | 0.846 ± 0.067 | 0.746 ± 0.030 | 0.641 ± 0.039 | 0.657 ± 0.110 | 0.577 ± 0.024 | 0.726 ± 0.049 | 0.707 |
| TabNet | 0.837 ± 0.004 | 0.672 ± 0.019 | 0.855 ± 0.110 | 0.754 ± 0.011 | 0.644 ± 0.038 | 0.715 ± 0.025 | 0.562 ± 0.045 | 0.562 ± 0.051 | 0.700 |
| Ours | **0.853 ± 0.003** | **0.702 ± 0.002** | **0.937 ± 0.011** | 0.761 ± 0.005 | **0.686 ± 0.007** | **0.745 ± 0.002** | 0.612 ± 0.008 | **0.912 ± 0.004** | **0.776** |
| Ours + Texts | — | — | — | — | — | — | 0.662 ± 0.006 | 0.967 ± 0.008 | **0.789** |

| Methods | Multi-Class Classification (F1 Micro Score) ↑ | | | Mean ↑ | Regression (MAE) ↓ | | | | Mean ↓ |
|---|---|---|---|---|---|---|---|---|---|
| | Thumbs-Up | Healthcare | Body Performance | | Sales | Stock | Air Quality | Employee Tenure | |
| XGBoost | 0.414 ± 0.000 | 0.344 ± 0.000 | 0.745 ± 0.000 | 0.501 | 0.0108 ± 0.0000 | 0.0322 ± 0.0000 | 0.0451 ± 0.0000 | **0.0717 ± 0.0000** | 0.0400 |
| RF | 0.410 ± 0.002 | **0.354 ± 0.008** | 0.747 ± 0.005 | 0.504 | 0.0110 ± 0.0001 | 0.0299 ± 0.0004 | 0.0166 ± 0.0008 | 0.0772 ± 0.0014 | 0.0337 |
| Logistics | 0.295 ± 0.000 | 0.324 ± 0.000 | 0.597 ± 0.000 | 0.405 | 0.0116 ± 0.0000 | 0.0345 ± 0.0000 | 0.0932 ± 0.0000 | 0.0803 ± 0.0000 | 0.0549 |
| KNN | 0.242 ± 0.000 | 0.335 ± 0.000 | 0.559 ± 0.000 | 0.379 | 0.0104 ± 0.0000 | 0.0275 ± 0.0000 | 0.0522 ± 0.0000 | 0.1118 ± 0.0000 | 0.0505 |
| MLP | 0.200 ± 0.000 | 0.332 ± 0.009 | 0.596 ± 0.070 | 0.376 | 0.0315 ± 0.0021 | 0.0291 ± 0.0014 | 0.0596 ± 0.0043 | 0.1202 ± 0.0070 | 0.0601 |
| TabNet | 0.401 ± 0.008 | 0.338 ± 0.007 | 0.743 ± 0.012 | 0.494 | 0.0115 ± 0.0010 | 0.0271 ± 0.0004 | 0.0611 ± 0.0046 | 0.0890 ± 0.0124 | 0.0472 |
| Ours | **0.451 ± 0.001** | 0.345 ± 0.000 | **0.752 ± 0.014** | 0.516 | **0.0071 ± 0.0002** | **0.0261 ± 0.0006** | **0.0140 ± 0.0048** | 0.0776 ± 0.0023 | **0.0312** |
| Ours + Texts | **0.480 ± 0.002** | — | — | 0.526 | — | — | — | — | — |

## 4.2 Implementation Details

We use LLaMA variant models as the pretrained LLM. For medical related datasets (as shown in Table 1), we use Bio-Medical-LLAMA-3-8B Con (2024). For the remaining dataset we use Meta-LLAMA-3-8B Dubey et al. (2024). The default hidden size of the transformer layer is set to 1024 while the default hidden size of the MLP layer is set to 512. The default learning rate is set to 1e-5 and the default weight decay is set to 1e-5. The total number of training epochs is 100, and the learning rate decreases by 10% every 10 epochs.

Among baseline methods, we refer to scikit-learn for the implementations of tabular prediction methods, including XGBoost, Random Forest (RF), Logistic Regression and K-nearest neighbors (KNN). We use the default hyperparameters and missing values are imputed using the mean value of each feature. A two-layer MLP serves as a baseline to verify the capacity of encoding in our method. Tabnet Arik & Pfister (2021) is based on pytorch-tabnet APIs, which is widely used in various works.

To ensure the statistical significance and robustness of our experiments, we conducted all the experiments five times and reported the results with 95% confidence intervals. We use the area under the receiver operating characteristic curve (AUC) Huang & Ling (2005) to evaluate binary classification; the F1-Micro Score Lipton et al. (2014) for multi-class classification; and the MAE Score Willmott & Matsuura (2005) for regression tasks. All experiments were conducted on an Nvidia A-100 GPU.

## 4.3 Experimental Results

In this section, we present the performance of ZET-LLM compared to other tabular prediction methods in Table 2, focusing on various tasks such as binary classification, multi-class classification, and regression. Since binary classification is the most common task in tabular prediction, many works primarily compare their methods on these tasks. To ensure a fair comparison, we evaluate all models without text features, as competing methods cannot process textual data. On binary classification tasks, ZET-LLM performs the best on 6 out of 8 datasets, outperforming both tree-based and deep learning methods. When considering the average performance across all binary classification datasets, ZET-LLM shows significant improvement over its competitors.In multi-class classification, ZET-LLM achieves the best results on 2 out of 3 datasets, while in regression tasks, it performs best on 3 out of 4 datasets. Overall, ZET-LLM consistently shows the best average performance across all task types.

Table 3: Ablation on different serialization methods (AUC).

| Input Form | SAD | Mortality | Decompensation | Respiration | Sepsis | Shock | IVF Pregnancy | Mean |
|---|---|---|---|---|---|---|---|---|
| Sample-Wise | 0.742 ± 0.004 | 0.645 ± 0.003 | 0.863 ± 0.007 | 0.755 ± 0.002 | 0.651 ± 0.003 | 0.710 ± 0.001 | 0.609 ± 0.013 | 0.711 |
| Feature-Wise | **0.853 ± 0.003** | **0.702 ± 0.002** | **0.937 ± 0.011** | **0.761 ± 0.005** | **0.686 ± 0.007** | **0.745 ± 0.002** | **0.662 ± 0.006** | **0.764** |

Table 4: Ablation on missing value mask (AUC).

| | SAD | Mortality | Decompensation | Respiration | Sepsis | Shock | IVF Pregnancy | Mean |
|---|---|---|---|---|---|---|---|---|
| w/o mask | 0.848 ± 0.006 | 0.691 ± 0.003 | 0.923 ± 0.008 | 0.752 ± 0.007 | 0.662 ± 0.003 | 0.723 ± 0.007 | 0.661 ± 0.005 | 0.751 |
| w/ mask | **0.853 ± 0.003** | **0.702 ± 0.002** | **0.937 ± 0.011** | **0.761 ± 0.005** | **0.686 ± 0.007** | **0.745 ± 0.002** | 0.662 ± 0.006 | **0.764** |

Table 5: Ablation on different pre-trained LLMs (AUC).

| Models | Embedding | SAD | Mortality | Decompensation | Respiration | Sepsis | Shock | IVF Pregnancy | Mean |
|---|---|---|---|---|---|---|---|---|---|---|
| Sheared-Llama-1.3B | 2048 | 0.835 ± 0.008 | 0.693 ± 0.007 | 0.942 ± 0.008 | 0.755 ± 0.003 | **0.691 ± 0.006** | **0.746 ± 0.007** | 0.627 ± 0.011 | 0.756 |
| Mistral-7B | 4096 | 0.851 ± 0.002 | 0.695 ± 0.002 | 0.942 ± 0.004 | 0.751 ± 0.004 | 0.690 ± 0.004 | 0.745 ± 0.005 | 0.628 ± 0.007 | 0.757 |
| Llama-2-7B | 4096 | 0.838 ± 0.007 | 0.692 ± 0.002 | 0.944 ± 0.002 | 0.750 ± 0.013 | 0.672 ± 0.002 | 0.742 ± 0.004 | 0.635 ± 0.007 | 0.753 |
| Llama-3-8B | 4096 | 0.850 ± 0.004 | 0.698 ± 0.001 | **0.947 ± 0.005** | **0.772 ± 0.003** | 0.690 ± 0.006 | 0.740 ± 0.008 | 0.627 ± 0.008 | 0.761 |
| Bio-Medical-Llama-3-8B | 4096 | **0.853 ± 0.003** | **0.702 ± 0.002** | 0.937 ± 0.011 | 0.761 ± 0.005 | 0.686 ± 0.007 | 0.745 ± 0.002 | **0.662 ± 0.006** | **0.764** |

One of the key advantages of ZET-LLM is its ability to understand and incorporate text features, a capability that competing methods lack. When text features are added to the datasets, ZET-LLM significantly outperforms all competing methods, demonstrating a substantial margin of improvement. This highlights the unique strength of ZET-LLM in handling complex, mixed-modal tabular data. These results validate that zero-shot encoding with LLMs is a powerful and effective approach for tabular prediction across various tasks. Furthermore, the ability to integrate textual features further enhances performance, which is especially crucial in scenarios involving complex tabular data.

## 4.4 ABLATION STUDY

**Ablation on serialization**  We conducted an ablation study to compare two serialization methods: feature-wise and sample-wise serialization (Table 3). As discussed in the introduction, feature-wise serialization consistently outperforms sample-wise serialization across datasets, with the largest performance gap observed on the SAD dataset, which contains the highest number of features. This result demonstrates that feature-wise serialization is more effective, particularly when dealing with datasets with many features.

**Ablation on missing value mask**  We evaluate our method with and without the missing value mask as shown in Table 4. The approach with the missing value mask performs better by mitigating the noise introduced by missing data. The performance gain on the IVF pregnancy and SAD datasets is low compared to others, which is expected given their low missing value ratios. This indicates that the missing value mask is particularly effective when the missing value ratio is higher. Furthermore, by simply masking missing values, we avoid the need for imputation.

**Ablation on different pre-trained LLMs**  To evaluate the performance of ZET-LLM across different pre-trained LLMs, we test it with various models, including Sheared-Llama-1.3B Xia et al. (2023), Mistral-7B Jiang et al. (2023), and Llama-2-7B Touvron et al. (2023a), on several datasets. The results, shown in Table 5, lead to two key conclusions.

First, the overall performance does not vary significantly with the size of the pre-trained LLM. The mean values across the different models are quite similar, with Sheared-Llama, despite having only 1.3B parameters and an embedding size of 2048, performing comparably to much larger models.

Second, pre-training on domain-specific data improves performance when domain knowledge is required. Although most medical datasets show minimal performance differences between Bio-Medical-Llama-3 and the other models, this is primarily because these datasets lack textual features. However, Bio-Medical-Llama-3 significantly outperforms other LLMs on the IVF Pregnancy dataset. This is because IVF contains doctors' comments as an important feature requiring medical domain knowledge to understand them better.

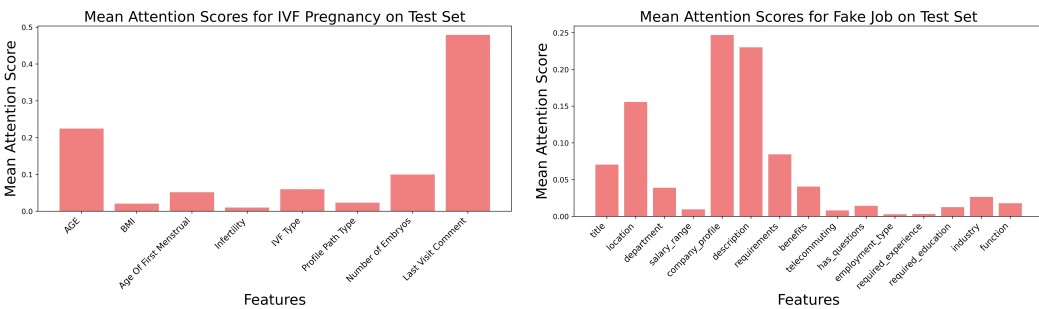

Figure 4: Left: mean attention scores for IVF Pregnancy on the test set. Right: mean attention scores for Fake job on the test set.

### 4.5 Interpretability Ananysis

We visualize the attention scores on the IVF Pregnancy and Fake Job test sets in Figure 4. For IVF Pregnancy, the attention score of "Last Visit Comments" is much higher than that of other features, which demonstrates the significance of textual information in this task. This observation is supported by the results in Table 2, where adding the textual feature increases the AUC from 0.612 to 0.662. Beyond the textual feature, "Age" has the second-highest attention score, indicating that our method considers Age a critical feature in predicting IVF outcomes. This finding is consistent with previous studies Sneed et al. (2008); Ubaldi et al. (2019), which emphasize the relevance of age in IVF success rates. For the Fake Job dataset, the textual features "company profile" and "description" achieve the highest attention scores, both of which are textual features. This confirms the importance of textual features. In addition to textual features, the categorical feature "location" is also crucial in prediction.

## 5 Discussion

**Conclusion**  In this paper, we introduced ZET-LLM, a novel approach that utilizes large language models (LLMs) as zero-shot feature extractors for tabular prediction tasks. Instead of fine-tuning pre-trained LLMs, we fine-tune only the task-specific modules, ensuring computational efficiency. We proposed a feature-wise serialization method, where individual features are encoded with equal emphasis on their semantic information. Additionally, a simple missing value masking mechanism effectively handles datasets with high missing value ratios. Altogether, ZET-LLM consistently outperforms traditional tabular models across binary classification, multi-class classification, and regression tasks. The method shows particularly significant gains when integrating textual features, offering a key advantage over models that are unable to process unstructured text.

**Future work**  In this work, our aim is to showcase the inherent potential of LLMs in tabular prediction without relying on complex, task-specific optimizations. While ZET-LLM demonstrates robustness in its current form, there are several promising directions for further improvement and extension of this method.

Prior research Slack & Singh (2023) has shown that task-specific instruction design can significantly enhance the performance of LLMs. Incorporating well-crafted, context-aware instructions could further optimize ZET-LLM. A potential future direction would be to explore how different types of instructions can be serialized and integrated with feature tokens for tabular data.

While we have shown that pre-trained LLMs can be effectively utilized for tabular prediction without fine-tuning, there is room to explore the benefits of adapting LLMs using parameter-efficient fine-tuning (PEFT) methods. Techniques such as LoRA Hu et al. (2022) or adapters Houlsby et al. (2019) could provide a lightweight way to fine-tune specific layers or modules of the model, allowing ZET-LLM to further adapt to domain-specific tasks with minimal computational overhead.

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

## A  DETAILS OF SOME DATASETS

- **Sepsis-Associated Delirium (SAD)** is a complex clinical syndrome, which is strongly associated with poor prognosis and long-term cognitive dysfunction. This task is based on Zhang et al. (2023b) and predicts whether or not patients have SAD based on tabular information from MIMIC-IV, such as initial vital signs and the use of mechanical ventilation.

- **Mortality:** This binary-classification task predicts in-hospital mortality based on observations recorded during an ICU admission. The original task from Harutyunyan et al. (2019) is formulated as a time-series classification task; we only use the most recent measurements of each stay for our paper.

- **Decompensation** refers to the rapid deterioration of patients' systems during their stay. The original task from is from Harutyunyan et al. (2019) contains millions of samples and is formulated as a time-series classification task. We sample $20,000$ stays from the task datasets and only use the most recent measurements for our model.

- **Respiration**, **Sepsis**, and **Shock** are common criticial conditions among adult ICU patients. We sample these binary classification tasks from phenotype classification in Harutyunyan et al. (2019).

- **In-Vitro Fertilization (IVF) Pregnancy** is based on a private, clinical dataset from the Tel Aviv Sourasky Medical Center. The pregnancy outcome is predicted using EHR data and doctors' comments.

- **Fake Job** is a dataset on Hugging Face that contains job postings labeled as either real or fake, aimed at detecting fraudulent job advertisements. It includes various features such as job titles, company names, and job descriptions, allowing for fraud detection.

- **Thumbs-Up** was used for training the PPrior Fereidouni et al. (2022). It is provided by RecmeApp and consists of user feedback data in the form of "thumbs up" or "thumbs down" ratings for items such as movies or products.

# B AVAILABLE VALUE RATIOS ACROSS DIFFERENT FEATURES

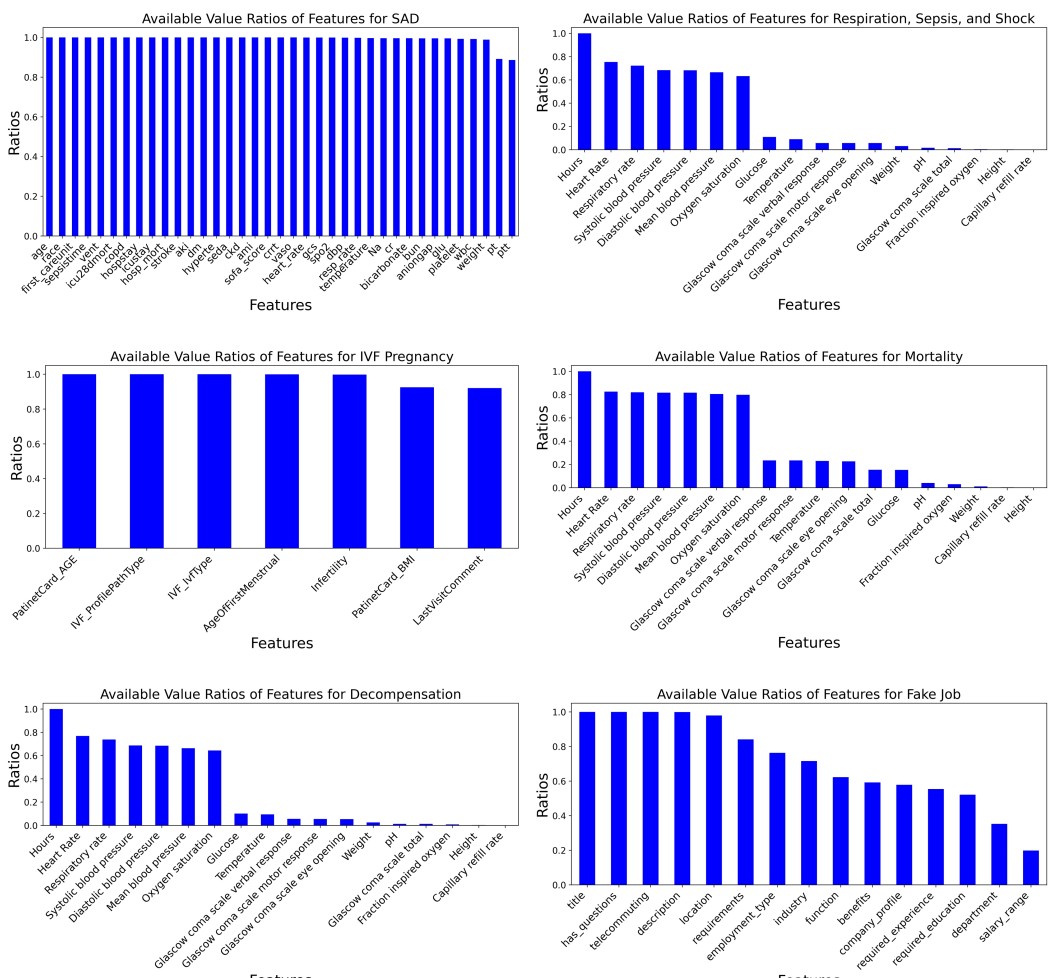

Figure 5: Available value ratios across different features in different datasets.

## C  MISSING VALUE DISTRIBUTION

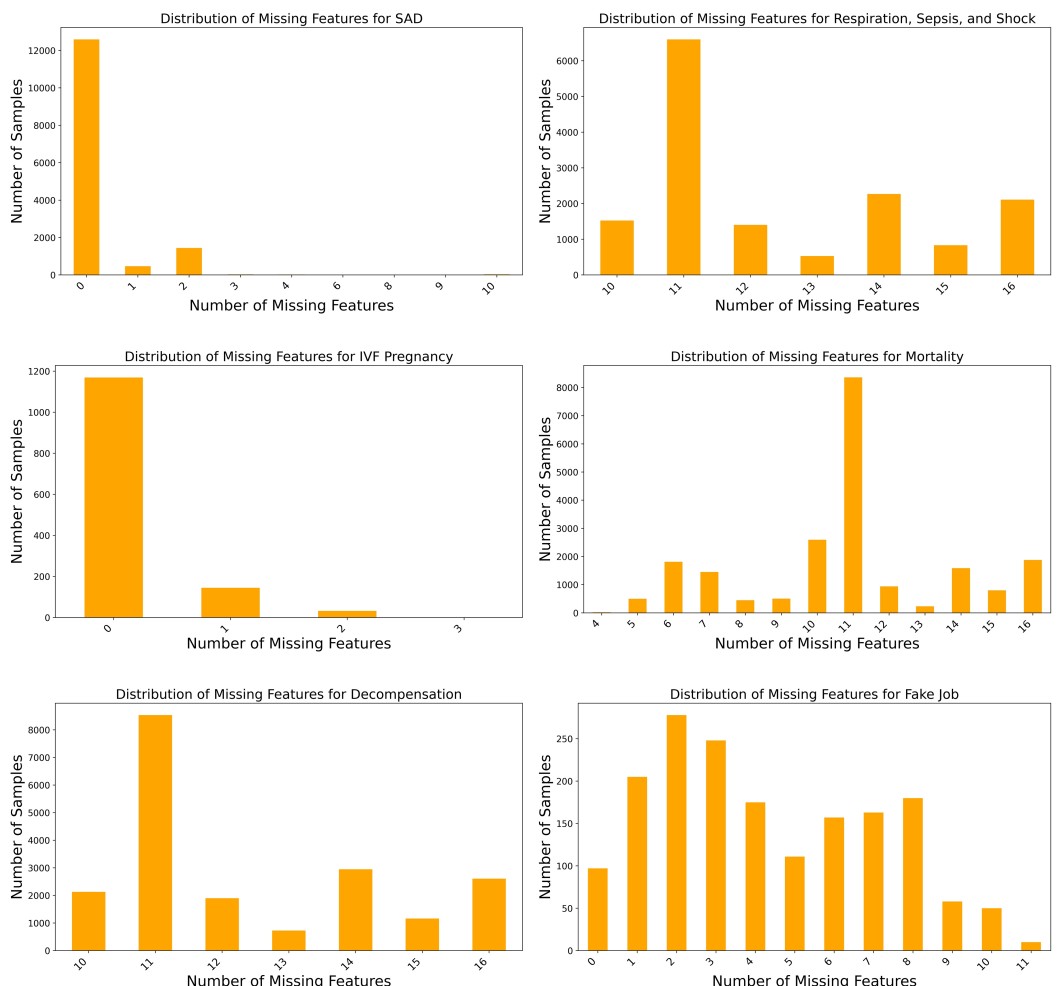

Figure 6: Distributions of missing features for different datasets.

# D ATTENTION SCORES FOR BINARY CLASSIFICATION DATASETS

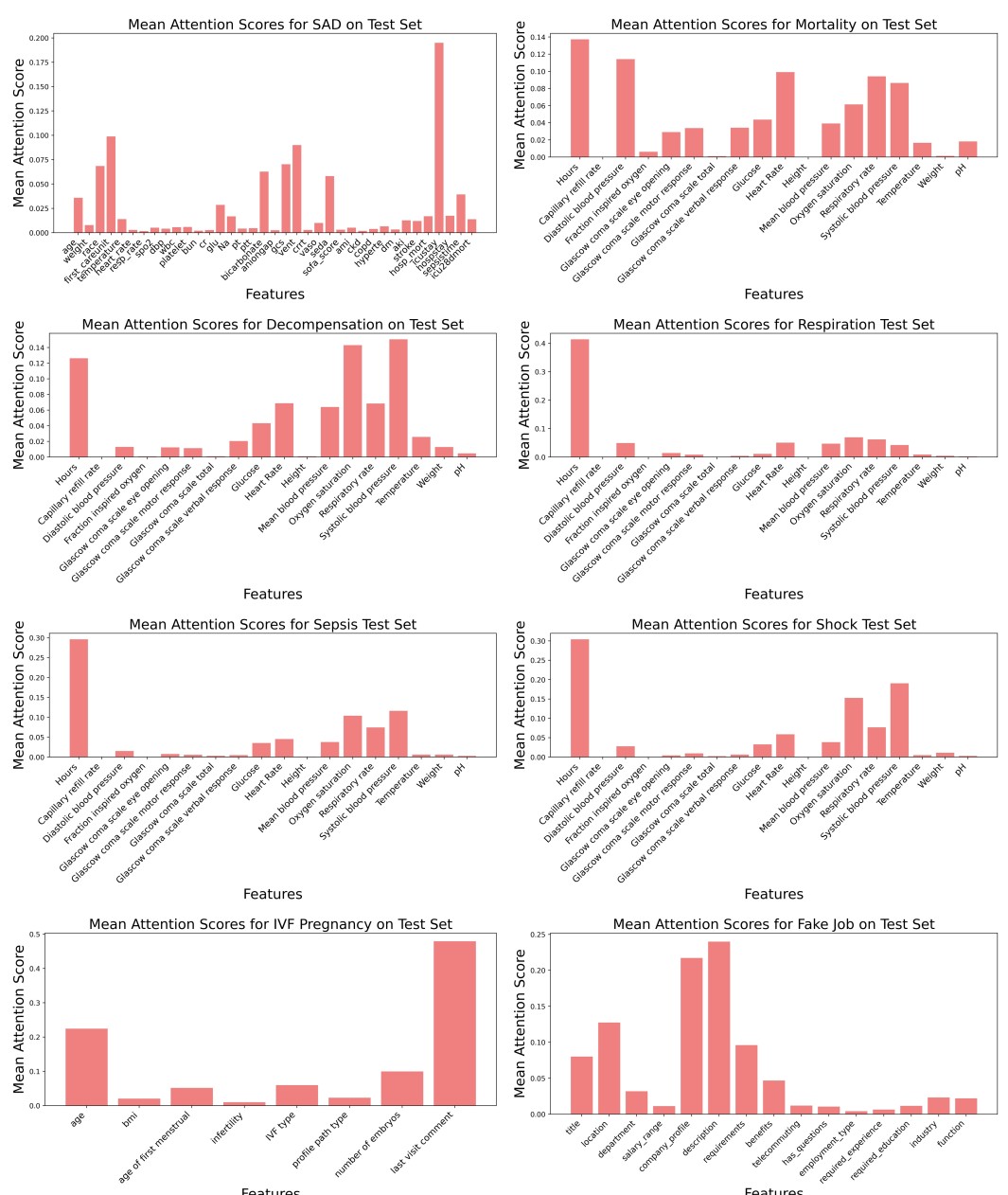

Figure 7: Mean attention scores for binary classification datasets on the test set.

# E ATTENTION SCORES FOR MULTI-CLASS CLASSIFICATION AND REGRESSION DATASETS

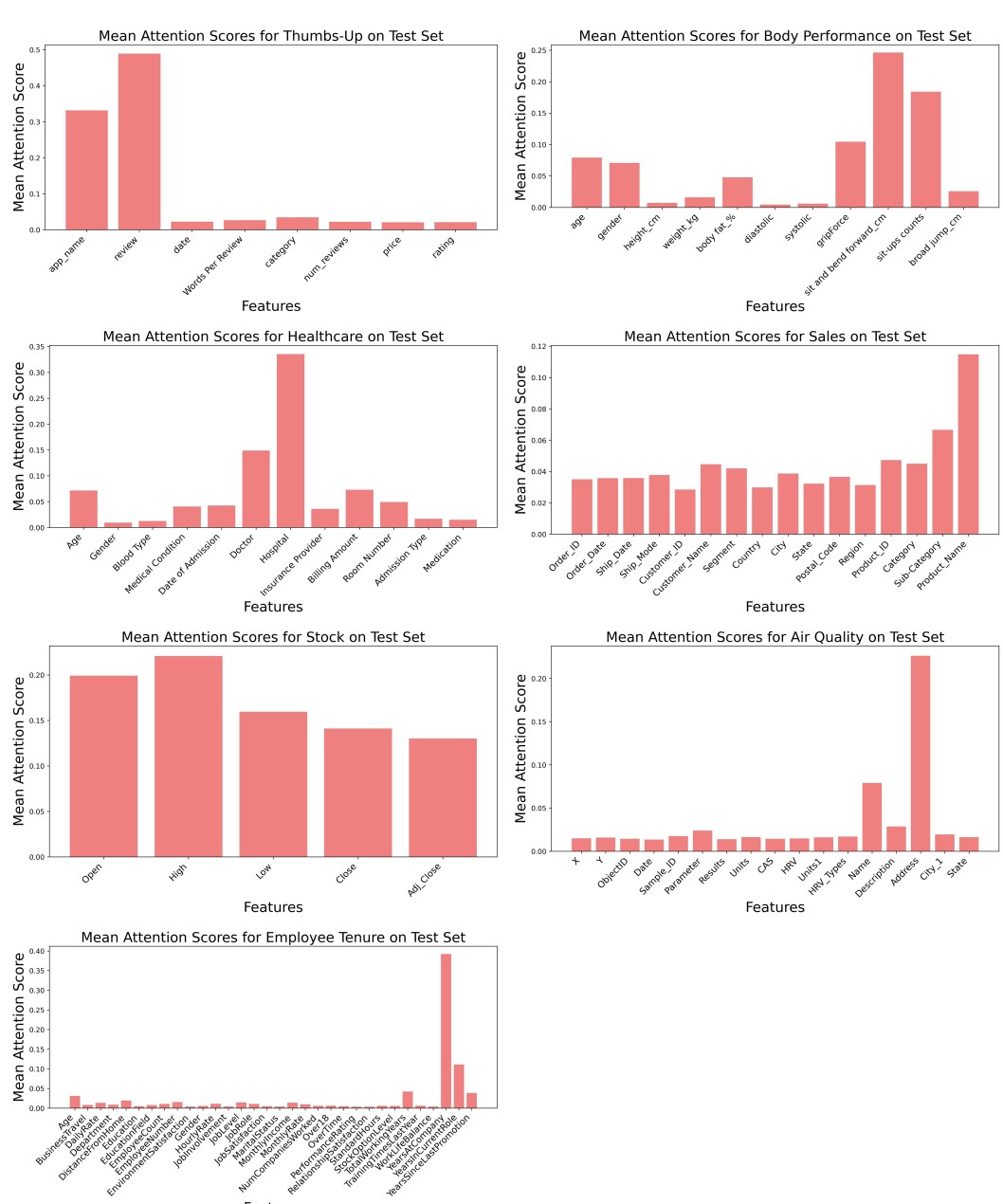

Figure 8: Mean attention scores for multi-class classification and regression datasets on the test set.

