# OpenReview forum: "Surprisingly Simple: Large Language Models are Zero-Shot Feature Extractors for Tabular and Text Data"
_ICLR.cc/2025/Conference — ICLR 2025 Conference Withdrawn Submission_

### Official Review · Reviewer_rtZk · 2024-10-30

**Soundness:** 2
**Presentation:** 3
**Contribution:** 2
**Rating:** 3
**Confidence:** 4

**Summary:**

- The paper leverages pre-trained LLMs as zero-shot feature extractors for tabular prediction tasks.
- Perform tabular-to-text serialization by encoding each feature as a separate token rather than representing the entire sample as a single token (contrary to TabLLM). They aim to treat each feature individually, ensuring that the model attends to all features more uniformly, preserving the order-invariance property.
- Replace the LLM’s autoregressive masking with bidirectional attention: enables the model to consider the entire context of a sentence at once. Then, they encode the serialized data into a sequence of token embeddings — token 1 = Age, token 2 = is, token 3 = 50. Lastly, they generate a feature token $z_i$ for each serialized sentence $t_i$ by applying mean pooling — $z$ = mean_pool(age, is, 50).
    - One sample: $\boldsymbol{z} = \{ z_i \mid i = 1, \ldots, n \}, \quad z_i = P(f_{\text{LLM}}(t_i))$
- Note on Missing Value Masking: If values for some features are missing i.e. Education value is missing, do not drop the entire sample, but just ignore it and perform the prediction per usual without the Education feature.
- Encoded Feature Token Aggregation: Combine all $z$’s of one sample using a trainable class $z_{cls}$ token.
    - Employ transformer blocks as the Feature Integrator to combine the individual feature tokens
    - The Feature Integrator learns to merge the information from each feature token into the class token, representing the full data sample
    - $h = f_{F1}(\boldsymbol{z}, z_{cls})$
- Prediction Layer: Leverages an MLP to perform predictions

**Strengths:**

**Originality**

Unique combination of feature-wise serialization and bidirectional attention in LLMs for tabular data.

**Quality**

Leverages a wide range of datasets but lacks a few key benchmarks such as Adult, Bank, Default etc. See Weaknesses.

**Clarity**

Paper is clear and well-written.

**Significance**

Results are promising, especially in its ability to understand and incorporate text features.

**Weaknesses:**

- L187-188: $x=\{(k_i, v_i)|i=1\ldots,n\}$ where $n$ denotes the number of features, $k$ and $v$ denote feature-name and value-pairs should be defined more clearly. I understand that you are trying to refer to a specific feature having a single possible value i.e. age = 50. However, an alternative way of interpreting this is where you can have $n$ features for $k_i$ i.e. age, education etc. but there can be more than $n$ different value-pairs. For example, for age where age = {1,2,3,…} where len(age) > $n$.
- L199-208: No citations of relevant papers i.e. justification for “LLMs often attend more strongly to tokens at the end of a sequence”.
- L235: There could be different approaches to pooling the tokens. For instance, why is it that mean pooling works? What about other pooling strategies?
- L300: Although the datasets are extensive, I would like to inquire regarding results and ablations on the most popular datasets such as the UCI [Adult](https://archive.ics.uci.edu/dataset/2/adult), [Bank](https://archive.ics.uci.edu/dataset/222/bank+marketing), and [Default](https://archive.ics.uci.edu/dataset/350/default+of+credit+card+clients) datasets.
- L368: Baselines from recent SOTA methods are missing. This includes the mentioned [TabLLM](https://arxiv.org/abs/2210.10723), and other methods such as [TabTransformer](https://arxiv.org/abs/2012.06678), [InterpreTabNet](https://arxiv.org/abs/2406.00426), [SAINT](https://arxiv.org/abs/2106.01342), [TabPFN](https://arxiv.org/abs/2207.01848) etc.
- L460: It is unclear how the ablation is conducted. What is the baseline model that the serialization is applied to? If it is ZET-LLM, would you please elaborate on the whole ablation process?
    - Although “feature-wise serialization consistently outperforms sample-wise serialization”, can I clarify that this can only be applied to your framework where you are required to first encode text into embeddings?
- L467: For w/o mask results, does that mean that you drop the samples instead of ignoring the masked features? On the other hand, in cases where missing values convey implicit information (e.g., non-response bias), could masking be suboptimal compared to other techniques, such as imputation or attention-based weighting?
- Given the adaptation of a transformer-based model for tabular data, is there an assessment of computational efficiency compared to traditional tabular models?

**Questions:**

It is unclear what the overall motivation of the paper is. Please see the weaknesses.

---

### Official Review · Reviewer_pHSk · 2024-11-02

**Soundness:** 2
**Presentation:** 2
**Contribution:** 1
**Rating:** 3
**Confidence:** 4

**Summary:**

This paper introduces a new method which uses LLMs as zero-shot feature extractors for tabular prediction tasks. The authors address token length limits by encoding each feature as a single token, which then forms a complete sample representation. It also incorporates missing value masking to handle gaps in complex datasets. This approach allows LLMs to process both structured and unstructured data effectively without fine-tuning or extensive preprocessing.

**Strengths:**

The paper raises an important problem on some issues with current input formats for table data to these LLMs and the issues with zero shot inference. The paper proposes to use LLMs to encodes rows to features and then use them to improve on specific tasks.,

**Weaknesses:**

> potential of autoregressive LLMs for tabular prediction has been explored only on a limited scale and with simpler datasets.

While there are a lot of research questions in this domain with LLMs that remain unexplored, this is a very poorly framed statement and does not make the paper any more convincing. There are tons of very strong research already conducted in this field with reasonably complex datasets.


The biggest weakness of this paper is that the ablations are too weak to conclude that feature generation is the desired way and that it alleviates problems with directly table to text serialization and the recent work on diffusion models with tabular data. Without comparisons to these methods, the proposed method cannot be claimed to have solved the problems of self attention on text data. I strongly suggest comparing [1] and [2] to the proposed method without which the proposed method cannot be claimed to have solved the existing problems.

The other big concern I have is that the finetuning is done specific to each task which immediately makes it non-scalable for deployment as the power of these models lie not in their specificity but for their generality so I strongly suggest that the authors do some kind of study with domain specific finetuning instead of task specific with these LLMs.

[1] TabLLM: Few-shot Classification of Tabular Data with Large Language
Models
[2] TabDDPM: Modelling Tabular Data with Diffusion Models

**Questions:**

> However, fine-tuning large language models is
computationally expensive and involves hyperparameters with complex scheduling strategies.

I do not see why hpo and scheduling would not be present for parameter efficient finetuning techniques as well. This does not seem to be an advantage to your work as well.

The authors talked about numeric features being a challenge. How do the authors propose to handle floating values over and above just using them as text?

Also, what about tables with large number of rows or columns? What truncation are the authors using?

---

### Official Review · Reviewer_GEBB · 2024-11-03

**Soundness:** 2
**Presentation:** 3
**Contribution:** 2
**Rating:** 3
**Confidence:** 3

**Summary:**

The authors propose a zero-shot Encoder for tabular data with LLMs, leveraging pre-trained LLMs as zero-shot feature extractors for tabular prediction tasks. They replace autoregressive masking with bidirectional attention, introduce a feature-wise serialization, and apply missing value masking to handle missing data. The study exhibits that LLMs can serve as a zero-shot feature extractor.

**Strengths:**

1. The manuscript is clearly written and the illustrative figure clearly shows the proposed method
2. Performance comparison covers different tasks including binary classification, multiclass classification and regression.

**Weaknesses:**

1. Some sentences are misleading. For example, “Third, when tabular data contains numerous features, encoding all of them into a single token burdens the LLM with a large set of information”. Term here should be token embedding instead of a token. Misusage of token in lieu of token embedding appears across entire page.
2. Innovation is limited. Text serialization is widely used in adapting LLMs for tabular data domains [1-3]. There is an ablation study on serialization. However, the study only scratches the surface. [1] and [3] show serialization methods using manual templates, table-to-text, and LLM can have remarkably different effects. The results of varying serialization methods are needed to demonstrate the effectiveness of the current method.
3. The motivation for applying bidirectional attention is not clear. When applying text serialization, a table is converted to multiple sentences. The input to frozen pretrained LLM is essentially natural language sentences. The reason for using bidirectional attention is not apparent. BERT [4] is the pioneering work demonstrating the power of bidirectional attention. However, autoregressive models nowadays show a more competitive performance than BERT. Providing empirical evidence comparing their bidirectional attention approach to an autoregressive approach is beneficial.
4. The order in Equation 3 is inconsistent with Figure 1. In equation 3, the combined embedding starts with z. However, in Figure 1, the combined embedding starts with z_cls.
5. Authors claim a single transformer block is sufficient to contextualize the CLS token embedding. Generally, a single transformer block has a limited capacity to aggregate information. The ablation study of pooling all token embeddings to get the final embedding instead of CLS token embedding is needed to support the claim. Thus, I suggest the authors compare performance with different numbers of transformer blocks.
6. Table 1 and Figure 2 are not the authors' contributions; they should be moved to the appendix.
7. In Table 2, comparing baselines requires the number of trainable parameters, total parameters, and FLOPs. The authors used Bio-Medical-LLAMA-3-8B and LLAMA-3-8B. The pretrained LLMs have a vast number of parameters, while baselines have a much smaller model size. Even though pretrained LLMs are frozen, the computational costs are much higher than baseline methods. Plus, LLMs are pretrained using extensive corpus data. Despite using much more computational resources, the performance gain is limited. A more fair comparison baseline can be Tablellama [9]. Thus, discussing the trade-offs between model size, computational cost, and performance gains is great.
8. Some of the sentences are not convincing. For example, authors attribute the reason why LLMs are underexplored in tabular data prediction to an autoregressive model. I do not see the logic here. The main reasons are reported in many references: the heterogeneous nature of table feature spaces [5-8].
[1] Hegselmann, Stefan, et al. "Tabllm: Few-shot classification of tabular data with large language models." International Conference on Artificial Intelligence and Statistics. PMLR, 2023.
[2] Dinh, Tuan, et al. "Lift: Language-interfaced fine-tuning for non-language machine learning tasks." Advances in Neural Information Processing Systems 35 (2022): 11763-11784.
[3] Jaitly, Sukriti, et al. "Towards Better Serialization of Tabular Data for Few-shot Classification." arXiv preprint arXiv:2312.12464 (2023).
[4] Devlin, Jacob. "Bert: Pre-training of deep bidirectional transformers for language understanding." arXiv preprint arXiv:1810.04805 (2018).
[5] Beyazit, Ege, et al. "An inductive bias for tabular deep learning." Advances in Neural Information Processing Systems 36 (2024).
[6] Yan, Jiahuan, et al. "Making pre-trained language models great on tabular prediction." arXiv preprint arXiv:2403.01841 (2024).
[7] Borisov, Vadim, et al. "Deep neural networks and tabular data: A survey." IEEE transactions on neural networks and learning systems (2022).
[8] Yan, Jiahuan, et al. "T2g-former: organizing tabular features into relation graphs promotes heterogeneous feature interaction." Proceedings of the AAAI Conference on Artificial Intelligence. Vol. 37. No. 9. 2023.
[9] Zhang, Tianshu, et al. "Tablellama: Towards open large generalist models for tables." arXiv preprint arXiv:2311.09206 (2023).

**Questions:**

All my questions are in the weaknesses section.

---

### Official Review · Reviewer_4hNv · 2024-11-04

**Soundness:** 2
**Presentation:** 2
**Contribution:** 2
**Rating:** 5
**Confidence:** 4

**Summary:**

This paper aims to use LLMs as an embedding extractor to perform tabular data prediction and propose ZET-LLM. To obtain each feature embedding, the proposed model first converts the tabular feature into a textual sentence with the "feature is value" template. and then use the LLM2VEC trick with mean pooling to get the final single vector. After feeding all feature sentences into LLMs, ZET-LLM integrates all feature embeddings using a transformer layer, and then the MLP layers are adopted to predict the values. the parameter in transformer-based integrator and MLP-based predictor are trained via task-specific loss. Experiments show the improvements of the proposed model.

**Strengths:**

1) The idea of employing LLM2vec for tabular data representation learning is interesting.

2) Feeding each feature sentence instead of the feature paragraph into LLM somewhat alleviates the problem of feature ordering.

3) The masking strategy in LLM seems to be friendly to the problem of missing values in tabular data.

4) Experiments and ablations show the superiority of the proposed modules.

**Weaknesses:**

1) The title of this submission is confusing for readers. Given the fact that the proposed modules (integrator and predictor) are trained with labeled samples, I think this title is inaccurate. (ZET-LLM may require many tabular samples).

2) The proposed model feeds each feature sentence into the LLM, which can lead to high time costs, especially for tabular datasets with hundreds of features.

3) What are the real empirical improvements of feeding each feature sentence into LLM (rather than the feature paragraph)

4) Technically, what is the main contribution of ZET-LLM compared to LLM2Vec.

**Questions:**

Please see the above weaknesses section.

---

### Official Review · Reviewer_2Ed5 · 2024-11-09

**Soundness:** 2
**Presentation:** 2
**Contribution:** 2
**Rating:** 5
**Confidence:** 4

**Summary:**

The paper introduces ZET-LLM, a method for using Large Language Models (LLMs) as zero-shot feature extractors for tabular prediction tasks, including binary and multi-class classification and regression. Traditionally, LLMs are designed for text, and applying them to tabular data has been challenging due to the complexity of converting structured data into text and handling diverse data types. ZET-LLM addresses this by transforming tabular data into a text-based format through feature-wise serialization, where each feature in a dataset is individually represented as a token. This approach allows the LLM to focus on each feature independently, ensuring order invariance and reducing the biases often present in sequential token generation. In addition, ZET-LLM employs encoded feature token aggregation, using a transformer-based "Feature Integrator" to combine individual feature tokens into a cohesive representation. This enables each feature to contribute uniformly to the final prediction while maintaining interpretability through attention scores. Additionally, the model incorporates a missing value masking technique to address common issues in complex datasets with missing information, further improving model performance without extensive data preprocessing or fine-tuning.

Experiments demonstrate that ZET-LLM outperforms traditional models and other deep learning methods across various datasets, including medical records and e-commerce data, by leveraging both structured data and unstructured textual information. This approach shows a notable advantage when handling datasets with mixed-modal data, as ZET-LLM can integrate textual features naturally, outperforming models that cannot handle unstructured text. Furthermore, an ablation study confirms the effectiveness of feature-wise serialization, encoded feature token aggregation, and missing value masking, highlighting ZET-LLM's robustness and adaptability. This work suggests that LLMs, even without fine-tuning, can effectively perform on tabular data, offering a lightweight, efficient solution for complex predictive tasks across different domains.

**Strengths:**

This paper presents a few strengths that advance the application of Large Language Models (LLMs) for tabular data tasks. First, by introducing a zero-shot framework, ZET-LLM eliminates the need for complex, resource-intensive fine-tuning, making it computationally efficient and accessible for various tasks. The feature-wise serialization approach innovatively addresses the inherent order invariance of tabular data, enabling the model to process each feature independently and uniformly, reducing biases associated with token position in the sequence. Additionally, the encoded feature token aggregation, with a transformer-based Feature Integrator,"effectively combines individual feature tokens while maintaining interpretability, allowing insights into feature importance through attention scores. These strengths highlight ZET-LLM’s potential as an efficient, adaptable model for advancing predictive tasks in tabular data.

**Weaknesses:**

A key weakness of the paper lies in its limited novelty. While ZET-LLM proposes an effective method for applying Large Language Models to tabular data, much of its approach builds on well-established concepts. For instance, the feature integrator component, although useful, closely resembles the mechanism used in TabTransformer, which also leverages attention to capture feature interactions. Additionally, the use of masking to handle missing values is not particularly innovative, as it has been widely used in previous work for similar purposes. Beyond the feature-wise serialization approach, which does contribute a fresh perspective to handling order invariance in tabular data, the paper does not introduce many new techniques. Consequently, the impact of ZET-LLM’s contributions may be limited, as the novelty beyond serialization is somewhat incremental compared to existing models in the field.

**Questions:**

Why did the authors choose to modify LLMs to generate embeddings specifically for tabular data, instead of utilizing existing LLM-based embedding models that are already optimized for generating embeddings?

It may be valuable to test ZET-LLM against pre-trained LLM embedding models designed for general-purpose feature extraction. This could provide insights into whether modifying LLMs for embeddings adds significant benefits or if similar results could be achieved using these models out of the box. Comparing ZET-LLM’s modified embeddings approach with standard LLM embedding models could help clarify if the additional complexity is truly necessary for performance improvements, particularly in handling diverse tabular data features.

---

### Note · Authors · 2024-12-02

**Comment:**

We appreciate all the reviewers' comments, which are great references for us. We admit that there are still many shortcomings in this article, so we have decided to withdraw it for further revisions. Thanks again to everyone for time spent on this article.

**Withdrawal Confirmation:**

I have read and agree with the venue's withdrawal policy on behalf of myself and my co-authors.